# Location, Location, Location: A Pilot Study to Compare Electrical with Echocardiographic-Guided Targeting of Left Ventricular Lead Placement in Cardiac Resynchronisation Therapy

**DOI:** 10.3390/diagnostics14030299

**Published:** 2024-01-30

**Authors:** Panagiota A. Chousou, Rahul K. Chattopadhyay, Gareth D. K. Matthews, Vassilios S. Vassiliou, Peter J. Pugh

**Affiliations:** 1Norwich Medical School, University of East Anglia, Norwich NR4 7TJ, UK; 2Department of Cardiology, Addenbrookes Hospital, Cambridge University Hospitals NHS Foundation Trust, Cambridge CB2 0QQ, UK

**Keywords:** heart failure, cardiac resynchronisation therapy, left ventricular lead placement, QLV, radial strain echocardiography

## Abstract

**Introduction:** Cardiac resynchronisation therapy is ineffective in 30–40% of patients with heart failure with reduced ejection fraction. Targeting non-scarred myocardium by selecting the site of latest mechanical activation using echocardiography has been suggested to improve outcomes but at the cost of increased resource utilisation. The interval between the beginning of the QRS complex and the local LV lead electrogram (QLV) might represent an alternative electrical marker. **Aims:** To determine whether the site of latest myocardial electrical and mechanical activation are concordant. **Methods:** This was a single-centre, prospective pilot study, enrolling patients between March 2019 and June 2021. Patients underwent speckle-tracking echocardiography (STE) prior to CRT implantation. Intra-procedural QLV measurement and R-wave amplitude were performed in a blinded fashion at all accessible coronary sinus branches. Pearson’s correlation coefficient and Cohen’s Kappa coefficient were utilised for the comparison of electrical and echocardiographic parameters. **Results:** A total of 20 subjects had complete data sets. In 15, there was a concordance at the optimal site between the electrically targeted region and the mechanically targeted region; in four, the regions were adjacent (within one segment). There was discordance (≥2 segments away) in only one case between the two methods of targeting. There was a statistically significant increase in procedure time and fluoroscopy duration using the intraprocedural QLV strategy. There was no statistical correlation between the quantitative electrical and echocardiographic data. **Conclusions:** A QLV-guided approach to targeting LV lead placement appears to be a potential alternative to the established echocardiographic-guided technique. However, it is associated with prolonged fluoroscopy and overall procedure time.

## 1. Introduction

Cardiac resynchronization therapy (CRT) is now a well-established part of guideline-directed therapy for a proportion of patients with heart failure with reduced ejection fraction [1,2]. Despite this, it is estimated that 30–40% of patients are “non-responders” to CRT [3]. This non-response rate has been relatively constant throughout the history of CRT and exists independent of the outcome measure used to describe response. Daubert et al. demonstrated this non-response rate across studies reporting hard outcome measures, remodelling measures of the left ventricle, functional measures and clinical composite scores [4].

Addressing this non-response rate represents one of the main challenges in CRT research. Careful patient selection and post-implantation optimisation are essential [5]. Similarly, rational LV lead positioning is of great importance, which is the focus of this study.

Originally, the LV lead was typically placed in a non-apical lateral or posterolateral region. This region tends to be the area of latest activation in patients with LBBB, and therefore, it is unsurprising that this location is effective for most patients. However, work over the last decade has suggested that a more individualised approach to LV lead positioning can optimise outcomes.

The TARGET [6] and STARTER [7] studies were two randomised control trials that utilised speckle-tracking echocardiography to identify the area of latest mechanical activation. This was then used to optimise the LV lead placement. Patients randomised to an echo-targeted strategy had improved responder rates in both haemodynamic and clinical outcomes.

Pacing site scar also play a deleterious role in the site of LV lead placement, with CRT response rates in patients with transmural posterolateral scar significantly reduced compared with those without scar [8].

The demonstration that a combination of the latest mechanical activation and the presence of scar impacts the response rate to CRT has led to a variety of different imaging modalities being used for pre-implant optimisation. MRI [9], PET-CT [10] and mixed modalities [11,12] have all been utilised with varying success.

None of these imaging modalities have resulted in a definitive answer to optimal LV lead positioning. Moreover, associated technical challenges with image acquisition and interpretation, additional cost, service availability, and the need for more hospital visits limit widespread adoption. These limitations have driven interest in intra-procedural electrical LV lead optimisation.

Two parameters of interest are the intrinsic left ventricular delay (QLV), which may be used to identify the area of latest electrical activation, and sensed R-wave amplitude, which may reflect the presence of myocardial scar. In this pilot study, we sought to assess the feasibility of using these parameters to guide LV lead placement and compare their suggested target with one identified using a speckle-tracking echocardiography strategy.

## 2. Materials and Methods

### 2.1. Patient Population and Study Protocol

This was a single-centre, non-randomised study that enrolled patients between March 2019 and June 2021. A total of 22 patients with a class I heart failure indication for CRT implantation were recruited. All patients were in sinus rhythm, with an LBBB morphology ECG (QRS duration > 130 ms) and an LV ejection fraction (EF) < 35%. The study was approved by the local ethics committee, and the study protocol complied with the guidelines set out in the Declaration of Helsinki. All participants gave fully informed written consent, and the trial was registered on a national database (NCT03769272).

### 2.2. Baseline Assessment

All participants underwent baseline clinical assessment, including heart failure aetiology phenotyping; symptom assessment; chamber quantification using echocardiography; and pre-implantation electrocardiography. Transthoracic echocardiography was performed using a commercial machine (Vivid E95, General Electric Medical Systems, Horten, Norway) using a 3.5-MHz phased-array transducer. Image interpretation was performed by a British Society of Echocardiography (BSE)-accredited individual, according to BSE guidelines. The end-systolic volume and ejection fractions were calculated using Simpson’s biplane method. An ischaemic phenotype was defined by the presence of coronary artery stenosis in either CT or invasive coronary angiography.

### 2.3. Speckle-Tracking Radial Strain Echocardiography

Speckle-tracking 2D radial strain analysis was performed on the mid-LV short-axis image (at the level of the papillary muscles) if it was deemed to be of sufficient quality. The mid-LV image was recorded at a minimum of 40 Hz. Commercial software (EchoPAC version 206, General Electric Medical Systems, Horten, Norway) was used for strain analysis. The endocardial border was identified manually via a point-and-click technique, with the software generating a concentric outer ring that correlated with the epicardium. Time–strain curves (Figure 1) were generated via motion analysis of the natural acoustic markers. The site of the latest mechanical activation was defined as the latest peaking segment compared with the onset of QRS duration.

Strain amplitude < 10% was used to identify LV scar. This value is based on previous work by our group [13].

### 2.4. QLV Measurement

The QLV was measured during the implant procedure by a cardiac physiologist and a cardiologist who were blinded to the radial speckle-tracking echo results. As previously described, the QLV interval was measured when patients were in sinus rhythm, during lead sensing [14]. Measurements were performed at a sweep speed of 100 mm/s, using a commercially available pacing system analyser (Abbott Laboratories, Chicago, IL, USA). The start of the interval was defined by the initial deflection of the QRS complex on the surface electrocardiogram channel. The end of the interval was defined by the first large negative or positive deflection on the left ventricular lead electrogram channel (Figure 2).

The QLV was measured in all accessible coronary sinus branches when a stable lead position was achieved.

### 2.5. R-Wave Amplitude Measurement

The amplitude of the sensed R-wave was measured at each accessible coronary sinus branch. This was recorded in millivolts.

### 2.6. CRT Implantation

Transvenous LV lead placement was performed. Balloon occlusive venography in steep left anterior oblique and posterior–anterior orientations allowed for the identification of accessible coronary sinus branches (Figure 3). Once a stable LV lead position was achieved in each accessible branch, a recording of the QLV and R-wave amplitude was performed. The operator was informed of the site of latest mechanical activation as identified via radial strain echocardiography after all measurements were performed. The final LV lead location was aimed at the site of the latest mechanical activation, as is local departmental policy if there was an accessible coronary sinus branch. When this was not possible, an adjacent segment was used preferentially over a remote site.

The right ventricular lead was positioned in the RV apex. The right atrial lead was positioned in the right atrial appendage.

Procedure times, durations of fluoroscopic screening, radiation exposure and contrast utilisation were recorded for each case.

### 2.7. Statistical Analysis

Statistical analysis was performed using commercially available software (SPSS, IBM, version 28.0, SPSS Inc., Chicago, IL, USA). Continuous variables are expressed as mean and standard deviation. Categorical variables are presented as frequencies and percentages. A *t*-test was used for the comparison of the means of continuous data. To assess for correlation between continuous data, Pearson’s correlation coefficient was used. When data were ranked to identify the area of latest electrical and mechanical activation, percentage agreement and Cohen’s κ statistic were utilised.

## 3. Results

### 3.1. Baseline Characteristics

A total of 22 patients were recruited to the study, all meeting the class I ESC recommendation for a heart failure CRT device, being in sinus rhythm with an LBBB and with an EF ≤ 35%. The baseline characteristics of the population are summarised in Table 1.

Patients were predominantly male and undergoing CRT-P implantation, with a mean age of 73.2 ± 9.0 years. As with most contemporary CRT populations, there was a mix of ischaemic and non-ischaemic heart failure phenotypes. All patients were established on heart failure guideline-directed medical therapy. The parasternal short-axis images for two of the patients were of insufficient quality for radial strain analysis.

### 3.2. Implant Characteristics

Implant characteristics are summarised in Table 2. All 22 patients undergoing CRT implantation had a lateral vein branch, but only 8 patients had a posterior vein. The majority of patients ultimately had the LV lead positioned in the lateral vein, with no patients undergoing anterolateral or anterior vein implantation.

The majority of patients had either three or four branches available for implantation, with only one patient having a single branch and one patient having five branches. No patients suffered complications during attempts to access their coronary sinus branches.

During the study, the implantation of quadpole leads became standard practice, and this became the preferred approach for the majority of cases. One of the unipolar leads was utilised because it was the only lead that could be implanted in the selected vein, with the rest of the unipolar and bipolar implants occurring before the adoption of the quadpole leads.

The mean procedure team was 124.3 ± 47.4 min, with an average fluoroscopy time of 29.3 ± 20.2 min.

### 3.3. Comparison of Procedural Elements to Published Standards

Durations from our centre for standard CRT procedures were used as a comparator for procedural aspects. The procedure time using a QLV-based strategy was 124.3 ± 47.4 min versus 83.6 ± 36.2 min for the usual CRT implant (*p* < 0.0001). Fluoroscopy time was also increased (QLV fluoroscopy time of 29.3 ± 20.2 min versus usual fluoroscopy time of 19.88 ± 16.3 min, *p* = 0.0085). A total of nine of the procedures were performed by a senior device trainee. If only consultant-performed procedures are considered, the total procedure time and fluoroscopy time were 112.7 ± 49.2 min and 24.8 ± 22.4 min, respectively. While the procedure time was statistically significantly different between consultant-performed procedures and standard care (112.7 ± 49.2 min versus 83.6 ± 36.2 min, *p* = 0.0087), there was no statistically significant difference in the fluoroscopy time for consultant-performed QLV-based procedures versus standard procedures (24.8 ± 22.4 min versus 19.88 ± 16.3 min, *p* = 0.2509), although this is in the context of a smaller sample size.

### 3.4. Comparing QLV and Radial Strain for Identification of the Area of Latest Activation

The longest QLV measurement was used to identify the area of latest electrical activation. This was defined as anterior, anterolateral, lateral, posterolateral, or posterior. Radial LV strain data from speckle-tracking echocardiography were similarly used to identify the area of latest mechanical activation. The area of latest electrical activation was compared with the area of latest mechanical activation and defined as concordant (at the optimal site), adjacent (within one segment) or discordant (≥2 segments away).

In 15 cases, there was concordance between the electrically targeted region and the mechanically targeted region; in four, the regions were adjacent. In only one case was there discordance between the two methods of targeting.

Consideration of the agreement between these two approaches suggests a percentage agreement of 75% and a Cohen’s κ statistic of 0.476. This may be interpreted as a moderate-strength agreement between the two techniques [16]. In the five cases of disagreement, only in one case were the segments identified by the two techniques not adjacent to each other.

Across the 20 patients who had satisfactory echocardiograms for radial strain analysis, 67 different QLV measurements were made at different coronary sinus branches. There was a weak positive correlation (Figure 4) between R-wave amplitude and the time to latest strain peak, which was not statistically significant (r = 0.10, *p* = 0.40).

### 3.5. Comparing Sensed R-Wave Amplitude and Strain Amplitude Defined LV Scar

Across the 20 patients who had satisfactory echocardiograms for radial strain analysis, 67 different R-wave amplitudes were measured at different coronary sinus branches. There was a weak positive correlation between R-wave amplitude and strain amplitude, which was not statistically significant (r = 0.09, *p* = 0.46).

The strain amplitude data were subsequently dichotomised according to the 10% cut-off, values under which were felt to represent scar. R-wave amplitude in the <10% group was, on average, 2.0 mV lower compared with the >10% group but did not reach statistical significance (*p* = 0.37).

## 4. Discussion

### 4.1. Summary of Results

In this study, we have demonstrated the feasibility of a QLV-based strategy for LV lead placement optimisation in a small but conventional CRT population. While there is no statistical correlation between the quantitative values of the QLV and the time to latest peaking strain, there was moderate agreement between the two techniques for the identification of optimal LV lead placement. In most cases where the longest QLV and time to peak strain did not match, the identified segments were adjacent to each other.

The lack of statistical correlation between the absolute quantitative values is perhaps not surprising given the complex nature of electro-mechanical coupling, which would indeed at least partly be contributed to by the electrical behaviour of LV scar tissue.

Despite the small sample, there appears to be no significant correlation between sensed R-wave amplitude and LV strain or the presence of strain-defined scar. This result has previously been alluded to, with findings revealing that similar sensing and pacing thresholds are seen in the presence of scarring localised to the pacing tip [17].

The QLV-based strategy was associated with an increase in both procedural and fluoroscopy time, reflecting the need for electrical interrogation at multiple coronary sinus branches and the initial learning curve of this novel technique. This is emphasised by the finding that, when only consultant-performed procedures were considered, there was no statistically significant difference in the total or fluoroscopy time of the procedure.

### 4.2. The Evidence for Targeting the Area of Latest Electrical Activation

There has been growing interest in the utilisation of electrical parameters for LV lead placement. Retrospective analysis of the SMART-AV trial suggested that prolonged QLV was a predictor of superior response in terms of reduction in mitral regurgitation 6 months post-implant and improvement in echocardiographic remodelling measures at 3 months [18]. A similar retrospective analysis of a single-centre study’s CRT database over nine years also showed a similar association between prolonged QLVs and predicting heart failure hospitalisation and mortality [19].

The ENHANCE CRT study (CRT Implant strategy using the Longest Electrical Delay for Non-Left Bundle Branch Block Patients) was the first prospective randomised control trial to compare QLV against anatomical implant approaches [20]. While it did not demonstrate any difference in responder rate, it is worth emphasising that this study recruited patients with non-LBBB interventricular conduction delays. The ElectroCRT study (electrically vs. imaging-guided implant of the LV lead in cardiac resynchronization therapy) subsequently demonstrated the non-inferiority of QLV targeting compared with a mixed imaging modality strategy, but this was in a purely ischaemic phenotype and was powered only for a composite clinical outcome [21].

Alternative parameters such as measures of interventricular delay (RV-LV) and inter-lead electrical delay have shown promise as alternatives or adjuncts to QLV for LV lead placement [22,23]. Particular focus should be made on parameters such as the interventricular delay between paced RV and sensed LV complexes (RVp-LV), which may represent a useful parameter in selecting LV placement in device upgrade cases, where other parameters are harder to interpret.

One of the potential difficulties with intraprocedural electrical measures is the different analysers that are used by different device manufacturers. Indeed, a recent study suggested that it is not possible to obtain accurate QLV measurements using certain analysers given the significant delay between the ECG and EGM channels [24]. This was not an issue with the pacing system analyser used for this study. Another suggested limitation of the QLV is the fact that it is primarily measured intra-procedurally, rather than providing pre-implant information as with imaging techniques. Of course, this does have the advantage of providing real-time information and also benefits from only providing information about accessible coronary sinus branches. Moreover, a recent abstract suggested that the QLV can be accurately estimated from a pre-implant 12-lead ECG, potentially providing a solution to this suggested limitation [25].

### 4.3. Why Is Imaging-Based Targeting Insufficient?

The TARGET and STARTER studies provided RCT evidence for the benefit of speckle-tracking radial strain echo-guided LV lead placement optimisation [6,7]. The 2013 ESC pacing guidelines provide a IIB recommendation for the targeting of the LV lead at the site of the latest activated LV segment [26]. This recommendation was not expanded upon in the most recent iteration of the guidance [2]. The lack of consistency in the findings of imaging-driven LV lead optimization is one of the main reasons that an alternative focusing on the area of latest electrical activation is being investigated. For example, while also using radial strain echo as part of a multi-modality targeting strategy, Borgquist et al. were unable to demonstrate an improved response rate, despite a comparable methodology to the TARGET and STARTER studies [11]. It is possible that one of the reasons for this inconsistent result is the technical challenge of obtaining accurate radial strain measurements [27].

In our own cohort, two patients did not have sufficient-quality transthoracic echo images to undertake radial strain analysis, a proportion similar to that seen in the TARGET study. This highlights another issue with imaging-based strategies. Transthoracic echo quality is highly operator- and patient-dependent, and the utilization of sub-optimal images can give rise to inaccurate strain imaging. While MRI does not suffer from an image quality issue, the inability of some patients to tolerate the scan is a commonly encountered issue. Moreover, strain assessment in both echo and MRI requires specialist commercial software, which may not be universally available and is vendor-specific.

The COVID-19 pandemic has also put resource utilization into the spotlight because of long waiting lists. This is also apparent in cardiac imaging services, and it is unclear if all centres would be able to cope with the increased demand from imaging-guided strategies and whether the additional hospital attendance required to facilitate such a strategy would fit into existing pathways.

Ultimately, we feel that a QLV-based strategy may prove to be a beneficial approach in cases where speckle-tracking echocardiography is not technically possible because of inadequate imaging quality.

### 4.4. Coronary Sinus Anatomy

While not the primary objective of this study, the study cohort provided further information about coronary sinus anatomy during transvenous LV lead placement. Previous work by our group demonstrated that only approximately 50% of patients had a single suitable vein for LV lead placement [17]. This limitation prompted further work on non-transvenous approaches to LV lead placement.

Despite the small size of our cohort, only one patient had a single accessible coronary sinus branch, with the majority having three or four. This finding likely reflects improvements in guiding catheter and lead technology, allowing for previously inaccessible branches to be implanted in. This demonstrates the need for older study results to be considered in light of developing technologies.

### 4.5. Future Directions

This study adds to the evidence for LV lead placement in the area of latest electrical activation. However, similar to the Stephenson study [21], the impact of scar via electrical targeting could not be accounted for. Thus, there is an obvious role for a combination of targeting methods for the area of latest electrical activation, with imaging-based myocardial scar identification. With the greater availability of cardiac MRI and the fact that it is becoming more routinely used for heart failure phenotyping, a study combining MRI-based scar identification and QLV targeting could be appropriate.

Further improvements in lead placement technology are ongoing, with growing interest in magnetic steering and navigation. One of the strengths of QLV is that it is a flexible technique that is not limited to transvenous implantation approaches, and it can be used in epicardial lead placement, especially in cases of lead repositioning due to non-response.

### 4.6. Study Limitations

While the main purpose of this study was to demonstrate the feasibility of an electrical strategy for lead optimisation, the small sample size and single-centre nature of the study limit the ability to draw statistical inferences from the data.

Given time and practical constraints, it was not possible to measure the outcome data of these patients in any form. This would have been particularly useful for cases where the QLV and radial strain echo were non-concordant, as it would potentially allow for the identification of the “correct” area for LV pacing and, thus, shed further light on the debate between targeting the area of latest electrical or mechanical conduction.

## 5. Conclusions

In conclusion, we have demonstrated that a QLV-based implant strategy for the area of latest electrical activation may be a suitable alternative to echo-guided targeting, although the technique does appear to be associated with prolonged fluoroscopy and procedure duration. We did not identify a suitable electrical surrogate for LV scar. This study is a pilot for a fully powered randomised study assessing the utilisation of a combination of QLV- and imaging-based strategies in patients with a class I recommendation for CRT implantation for heart failure.

## Figures and Tables

**Figure 1 diagnostics-14-00299-f001:**
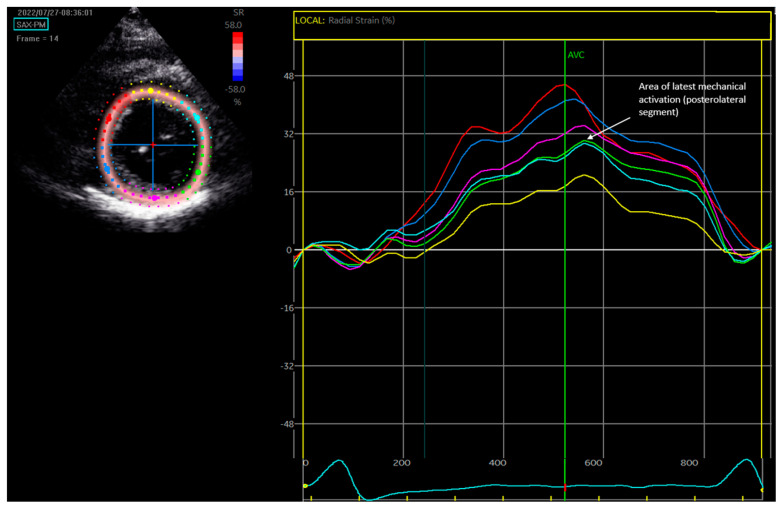
Speckle−tracking echocardiography to determine the region of the latest mechanical activation. **Top Left**: Mid-level LV parasternal short-axis transthoracic images from which the speckle-tracked traces are derived. **Right**: The time speckle-tracked traces. The latest segment of mechanical activation is the posterolateral wall (green line). Time to latest mechanical activation is measured as the difference from the start of the surface ECG QRS complex (vertical yellow line) to the latest peaking segment.

**Figure 2 diagnostics-14-00299-f002:**
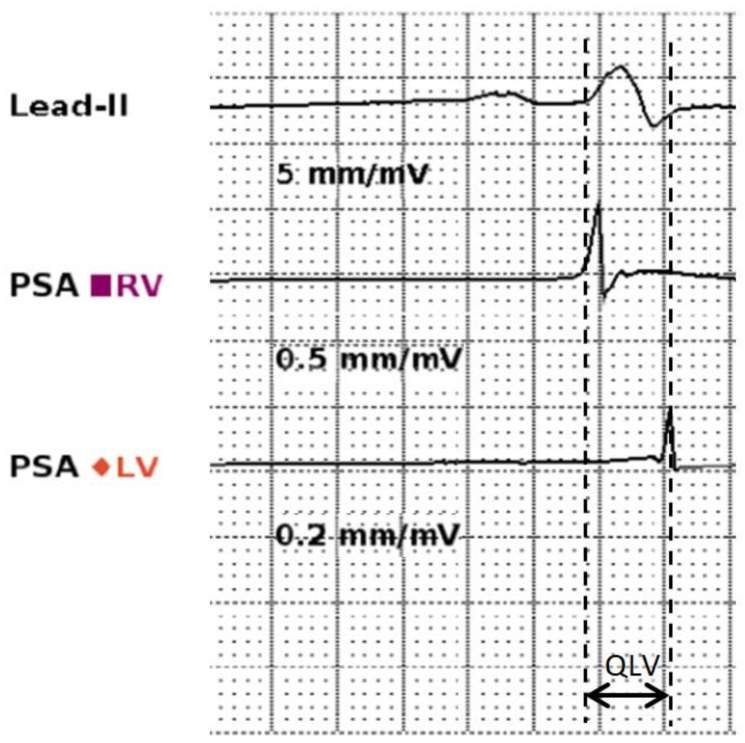
Example of QLV measurement. Measurements were taken from the pacing system analyser at 100 mm/s speed. The left dashed line represents the start of the QRS complex as seen on the surface electrocardiogram (top trace); the right dashed line represents the first major deflection of the LV electrogram. The QLV is marked by the double-headed arrow.

**Figure 3 diagnostics-14-00299-f003:**
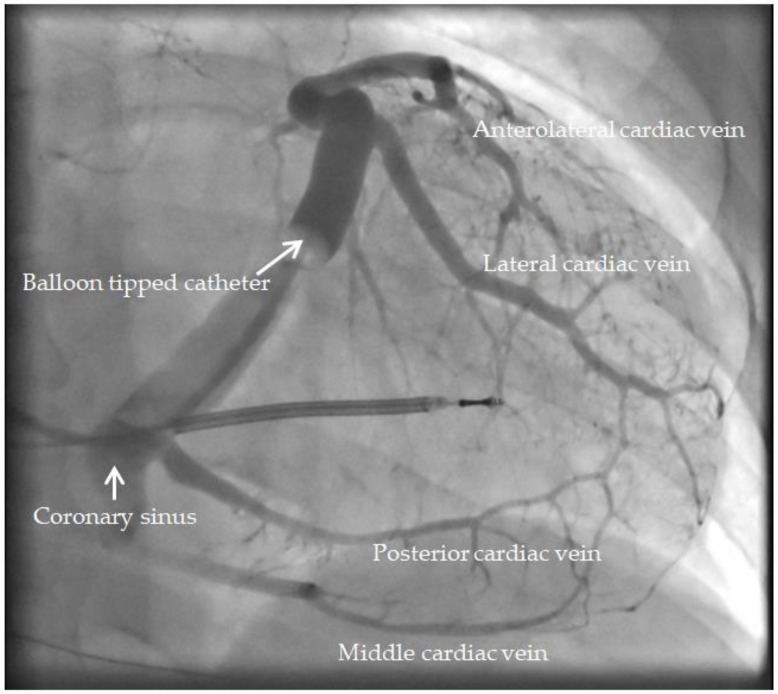
Balloon occlusive coronary sinus venogram, posterior-anterior projection. Image adapted from Nayar et al. [15], with permission.

**Figure 4 diagnostics-14-00299-f004:**
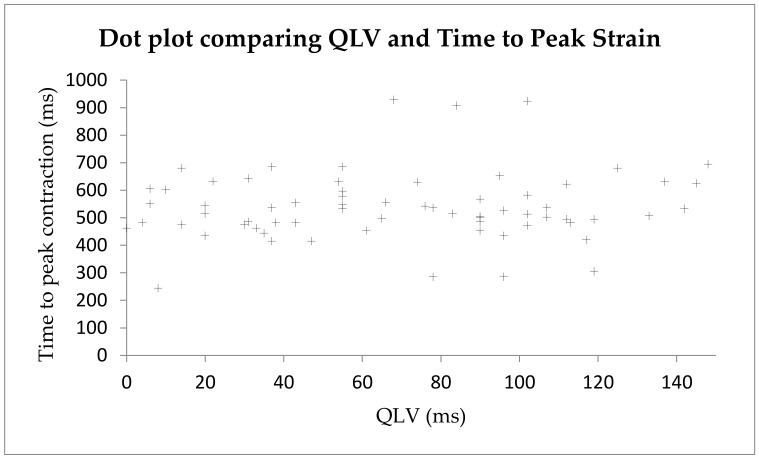
Dot plot showing a weak positive correlation between the QLV and time to peak strain.

**Table 1 diagnostics-14-00299-t001:** Baseline characteristics. Continuous variables are presented as mean ± standard deviation; ordinal variables are presented as a number (%).

	*n* = 22
*Patient Demographics*	
Age (years)	73.2 ± 9.00
Male	17 (77.2)
CRT-P	14 (63.6)
CRT-D	8 (36.4)
NYHA 1	0 (0)
NYHA 2	7 (31.8)
NYHA 3	13 (59.1)
NYHA 4	2 (9.1)
*Pre-implant ECG*	
QRS duration (msec)	159.8 ± 17.5
PR duration (msec)	186.2 ± 36.0
*Heart failure phenotype*	
Ischaemic	8 (36.4)
Non-ischaemic	14 (63.6)
*Pharmoctherapy*	
Beta-blocker	21 (95.5)
MRA	11 (50.0)
RAASi	21 (95.5)
ACEi	13 (59.1)
ARB	4 (18.2)
ARNi	4 (18.2)
Sufficient quality echo for strain analysis	19 (86.4)

CRT—cardiac resynchronisation therapy; NYHA—New York Health Association; ECG—electrocardiogram; MRA—mineralocorticoid receptor antagonist; RAASi—renin angiotensin aldosterone system inhibitor; ACEi—angiotensin-converting enzyme inhibitor; ARB—angiotensin receptor blocker; ARNi—angiotensin receptor/neprilysin inhibitor.

**Table 2 diagnostics-14-00299-t002:** Implant characteristics. Continuous variables are presented as mean ± standard deviation; ordinal variables are presented as a number (%).

	*n* = 22
*Presence of coronary sinus branch*	
Posterior vein	8 (36.4)
Posterolateral vein	11 (50.0)
Lateral vein	22 (100)
Anterolateral vein	14 (63.6)
Anterior vein	18 (81.8)
*Site of LV lead placement*	
Posterior vein	4 (18.2)
Posterolateral vein	4 (18.2)
Lateral vein	14 (63.6)
Anterolateral vein	0 (0)
Anterior vein	0 (0)
*Number of accessible coronary sinus branches*	
1	1 (4.54)
2	2 (9.09)
3	9 (40.9)
4	9 (40.9)
5	1 (4.54)
*Lead type*	
Unipolar	2 (9.09)
Bipolar	4 (18.2)
Quadripolar	16 (72.7)
Procedure time (minutes)	124.3 (47.4)
Contrast volume (mL)	54.3 (67.8)
Skin dose (mGy/m^2^)	330.1 (396.6)
Fluoroscopy time (minutes)	29.3 (20.2)
Number of procedures by senior trainee	9 (40.9)

## Data Availability

The data are available upon request. The data underlying this article will be shared upon reasonable request to the corresponding authors.

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
