# Peer review of "Location, Location, Location: A Pilot Study to Compare Electrical with Echocardiographic-Guided Targeting of Left Ventricular Lead Placement in Cardiac Resynchronisation Therapy"

_diagnostics, 2024, doi:10.3390/diagnostics14030299_

Round 1

Reviewer 1 Report

Comments and Suggestions for Authors

In order to optimize CRT procedure (cardiac resynchronization therapy), in a few patients scheduled for  CRT (22 patients) , the authors attempted to use a relatively new pure electrical marker and compare it with a most established mechanical marker attained by echocardiography (radial strain attained by angle independent speckle tracking technology). In particular they tried to measure the QLV (the time interval from the first deflection on a surface ECG to local intrinsic activation at the LV stimulation site).  They found a moderate agreement with radial strain activation (K statistics of 0.47) with a categorical approach  but the quantitave correlation was poor (r 0.1, p=0.4). Also to assess LV scar the R-wave amplitude correlated poorly with strain amplitude.

A certain criticism has to be manifested.

1)    The authors measured in different coronary sinus branches. Was there any complications during this attempts? This represents a possible drawback of this new approach further compounding the reported prolongation of the damaging fluoroscopy  time.

2)    The problem of the scar is totally overlooked by this new method from a theoretical standpoint also confirmed by the authors data. How do the authors intend to tackle this point that in terms of CRT efficacy is a major one? We do not think that a routine MRI could be a solution because MRI is also mutagenic so further expanding the genotoxic risk already exerted by the increased fluoroscopy time [1] . We think that in this respect ultrasounds are the best approach and only in few cases with poor speckle tracking data (limited window) other more risky approaches should be tried.

3)    The problem is that this parameter is attainable only during the cath. I am not sure if after the procedure with restoration of the normal autonomic tone this parameter could remain constant. This in my view is a major limitation of the study.

4)    They reported a value of K statistics of .47  that they say should indicate a moderate agreement. In reality in accordance with Peat (2001, p 228) only  a K value of at least 0.5  is moderate.  So there is less than a moderate agreement and furthermore no population inference can be done being the p> 0.05. That should be fixed in the text.

5)    Minor points:  QLV first use in text is without legend and  at the row 271 the numbering of the figure is wrong (figure 1 instead of figure 4 ).

6)              

7)              

8)             References

9)              

10)          1.         Fiechter M, Stehli J, Fuchs TA, Dougoud S, Gaemperli O, Kaufmann PA. Impact of cardiac magnetic resonance imaging on human lymphocyte DNA integrity. Eur Heart J. 2013;34(30):2340-5. Journal Article)           

Comments on the Quality of English Language

English is fine

Reviewer 2 Report

Comments and Suggestions for Authors

This is a nice study of 22 patients with HFrEF and LBBB submitted to CRT in whom concordance between intraprocedural LV electrical delay and pre procedural  LV mechanical delay was investigated (actually only 20 patients were analysed duo to lack of appropriate echocardiographical window). Although not original the study suggest that striving for the longest intraprocedural electrical LV delay will potentially add value in less than 25% of patients (in 75% of patients there is electro-mechanical concordance) with the price of prolonged procedure and X-ray exposure.

Although the paper is well written overall, I found as a major weakness the low quality of abstract (excessive introduction, lack of description of working protocol in the methods section).

Another strong suggestion for improvement is changing the conclusion in order to emphasise the increased procedure duration and X-ray exposure with this strategy.

Round 2

Reviewer 1 Report

Comments and Suggestions for Authors

The paper has been a little bit improved.

Reviewer 2 Report

Comments and Suggestions for Authors

The authors performed the required improvements.